# Site-specific analysis of N-glycans from different sheep prion strains

**Natali Nakić**[1,2☯], **Thanh Hoa Tran**[2,3☯], **Mislav Novokmet**[1], **Olivier Andreoletti**[4], **Gordan Lauc**[1,5], **Giuseppe Legname**[2,6]*

**1** Genos Glycoscience Research Laboratory, Zagreb, Croatia, **2** Laboratory of Prion Biology, Department of Neuroscience, Scuola Internazionale Superiore di Studi Avanzati (SISSA), Trieste, Italy, **3** VNUK Institute for Research and Executive Education, The University of Danang, Da Nang, Vietnam, **4** UMR INRA ENVT 1225-IHAP, École Nationale Vétérinaire de Toulouse, Toulouse, France, **5** Faculty of Pharmacy and Biochemistry, University of Zagreb, Zagreb, Croatia, **6** ELETTRA Sincrotrone Trieste S.C.p.A., Basovizza, Trieste, Italy

☯ These authors contributed equally to this work.
* legname@sissa.it

**Data Availability Statement:** All relevant data are within the manuscript.

**Funding:** The authors received no specific funding for this work.

## Abstract

Prion diseases are a group of neurodegenerative diseases affecting a wide range of mammalian species, including humans. During the course of the disease, the abnormally folded scrapie prion protein (PrP$^{Sc}$) accumulates in the central nervous system where it causes neurodegeneration. In prion disorders, the diverse spectrum of illnesses exists because of the presence of different isoforms of PrP$^{Sc}$ where they occupy distinct conformational states called strains. Strains are biochemically distinguished by a characteristic three-band immunoblot pattern, defined by differences in the occupancy of two glycosylation sites on the prion protein (PrP). Characterization of the exact N-glycan structures attached on either PrP$^{C}$ or PrP$^{Sc}$ is lacking. Here we report the characterization and comparison of N-glycans from two different sheep prion strains. PrP$^{Sc}$ from both strains was isolated from brain tissue and enzymatically digested with trypsin. By using liquid chromatography coupled to electrospray mass spectrometry, a site-specific analysis was performed. A total of 100 structures were detected on both glycosylation sites. The N-glycan profile was shown to be similar to the one on mouse PrP, however, with additional 40 structures reported. The results presented here show no major differences in glycan composition, suggesting that glycans may not be responsible for the differences in the two analyzed prion strains.

## Author summary

To date, prion diseases remain a controversy amongst scientists. Although we know now it is the abnormal form of the prion protein (PrP$^{Sc}$) that causes the disease, many questions are still left unanswered. To understand the cellular mechanism of these diseases, we should first and foremost try to fully understand the prion protein itself. Even though many findings have been made regarding the structure of the protein, a large part of it is still unknown. Since the prion protein is actually a glycoprotein, to resolve its structure we need to put our focus not only on the protein part of the glycoprotein but also on the glycan structures as well. Here we compared two different sheep prion strains and although

**Competing interests:** I have read the journal's policy and the authors of this manuscript have the following competing interests: GL is the founder and CEO of Genos – a private research organization that specializes in high-throughput glycomic analysis and has several patents in the field. NN and MN are employees of Genos. Other authors declare no competing financial interests.

no major differences have been found between the glycan structures, this analysis may help the understanding of the role glycans have in prion diseases.

## Introduction

Prion diseases, or transmissible spongiform encephalopathies (TSEs), are a group of fatal and infectious neurodegenerative diseases. These disorders are characterized by brain inflammation, spongiform degeneration, and accumulation of infectious prion protein denoted as scrapie prion, eventually leading to neuronal death [1,2].

The main event in the disease is the misfolding of a sialoglycoprotein, the normal cellular prion protein (PrP$^C$), to the pathological form (PrP$^{Sc}$). The two isoforms of the protein have the same primary amino acid sequence, with two N-glycosylation sites (at positions N-184 and N-200 in sheep) preserved in both forms. They differ in their secondary structure–while PrP$^C$ is rich in α-helices, PrP$^{Sc}$ has a high proportion of β-sheets [3,4]. Other main differences between the two forms are their solubility in non-denaturant detergents and partial resistance to proteinase K (PK). PrP$^C$ is PK-sensitive and completely degraded upon treatment with proteases, while PrP$^{Sc}$ is partially PK-resistant, leading to the formation of a resistant PrP$^{Sc}$ core, termed PrP$^{res}$. To biochemically distinguish the two forms, Western blotting them after PK treatment is routinely employed. PrP$^{res}$ (PK-resistant form of PrP) displays characteristic three-band pattern, corresponding to the diglycosylated (where both glycosylation sites are occupied), monoglycosylated (only one of the two sites is occupied), and unglycosylated band [5].

The prion concept, now widely accepted, was first proposed by Stanley B. Prusiner in 1982 [6], stating that prions are infectious proteins responsible for causing prion diseases. Many lines of evidence support this concept [7–9], with the main and final proof being the production of synthetic prions [10] in laboratory. However, one aspect of prion biology, the existence of strains, still needs to be fully elucidated. Strains are present in all prion diseases and differ by clinical symptoms, incubation periods, neuropathology, and biochemical properties, including electrophoretic mobility, glycosylation pattern (differing in occupation of glycosylation sites on the PrP itself), and partial, relative resistance to PK [11,12]. It is believed that strain differences exist due to distinct conformational states which PrP$^{Sc}$ is able to occupy [13,14]; replicated in a strain-specific manner within the host; proposed to be influenced by polymorphisms, and its glycosylation status [15,16]. Even though it is known that prion strains accumulate selectively in distinct brain regions, glycosylation of PrP does not seem to be necessary for this phenomenon to occur [17].

Glycosylation is a relevant co- and post-translational modification. N-glycans generally have a major role in affecting the structure, stability, and folding of proteins. In recent years, increasing attention has been put on investigating glycosylation of the prion protein. In prions, N-glycans might affect the rate of protein cellular trafficking, protein aggregation, and fibril formation [18,19]. Studies introducing point mutations in the *Prnp* gene, altering glycosylation patterns, together with the use of different cell lines and different constructs, often led to contradictory results. For instance, Rogers *et al.* (1990) used cell constructs with mutations in the conserved tripeptide sequence (specifically, in the threonine residue, T183A, and T199A in hamster) and showed that instead of correctly trafficking to the cell membrane, unglycosylated PrP was intracellularly accumulated [20]. Using the same construct, DeArmond *et al.* (1997) claimed that presumably because of this intracellular localization, PrP$^C$ is not able to convert to PrP$^{Sc}$ [21]. This led to the suggestion that N-glycans are necessary for the correct

localization of PrP. Korth *et al.* (2000) and Neuendorf *et al.* (2004) used different genetically modified mice expressing PrP mutated constructs (N180Q, N196Q, and T182N, T198A, respectively) and once again these studies showed different results. In the unglycosylated mutants, PrP$^C$ was localized on the cell surface and readily converted to PrP$^{Sc}$, thus indicating that glycans are actually not necessary for prion infection [22,23]. It was suggested that, rather than the lack of glycans, the point mutations themselves are the ones responsible for PrP properties changes, which lead to its incorrect trafficking [24]. Also, Cancellotti *et al.* (2005) showed that the levels of mutated constructs expressing the unglycosylated PrP$^C$ were comparable to the levels in wild-type PrP and that it was mainly accumulated intracellularly. The diglycosylated and monoglycosylated PrP, however, are localized on the cell surface, implying that glycans have a role in determining the localization of PrP [15,25]. All these results lead to vague conclusions and show that glycosylation of PrP needs to be investigated further.

Until now, only a few studies have dealt with structural analysis of N-glycans attached to PrP. Endo *et al.* (1989) isolated PrP$^{Sc}$ from Syrian hamster brains and analyzed released glycans, in combination with exoglycosidase digestion. The analysis revealed that mainly complex N-glycans (bi-, tri- and tetraantennary) are present on the two glycosylation sites [26]. Next, Rudd *et al.* (1999) did a comparison of released and labeled N-glycans from PrP$^C$ and PrP$^{Sc}$ where they managed to detect around 50 N-linked glycans in total. They concluded that the two forms carry the same set of N-glycans, but the relative proportion of each structure was different. They hypothesized that this difference could be because of changes in the activity of a specific glycosyltransferase in the infected brain [27]. The first detailed site-specific analysis of N-glycans was performed on PrP$^{Sc}$ isolated from infected mice brains (glycosylation sites at N-180 and N-196). Researchers analyzed PrP$^{Sc}$ glycopeptides and detected around 60 complex N-linked glycans, characterized as bi-, tri- and tetraantennary, fucosylated (with both core fucose and outer arm fucose) and sialylated (up to three sialic acid residues). The analysis also showed differences in the glycan composition between the two glycosylation sites–glycans at N-180 were mostly bi- and triantennary, while at N-196 there was a majority of tri- and tetra-antennary structures, sialylated to a higher degree than the glycans on the first glycosylation site [28].

Glycosylation as a post-translation modification leads to great complexity and variability of structures. The diversity of glycans arises from the different composition and sequence of monosaccharide units in the polymer and the configuration of glycosidic linkages between the units. A single glycosylation site can carry many different glycans, meaning that increasing the number of glycosylation sites causes a large microheterogeneity in a glycoprotein, making glycan analysis more challenging compared to other post-translational modifications. In this study, structural characterization and comparison of N-linked glycans from two different ovine prion strains have been performed. The method of choice was the analysis of PrP$^{Sc}$ glycopeptides since it leads to site-specific information. This study suggests that glycosylation may not be the factor influencing prion strains diversities, at least in the two strains analyzed in this study.

## Results

### PrP$^{Sc}$ isolation and purification

To perform structural characterization of N-glycans, the starting material should be as pure as possible [29]. Ever since Prusiner and his colleagues [30,31] purified the prion protein, many different approaches have been tested for prion protein isolation. However, there is still no straightforward method for isolating prions. Therefore, a large amount of brain tissue is necessary to optimize protein isolation. The current isolation was based on the protocol from

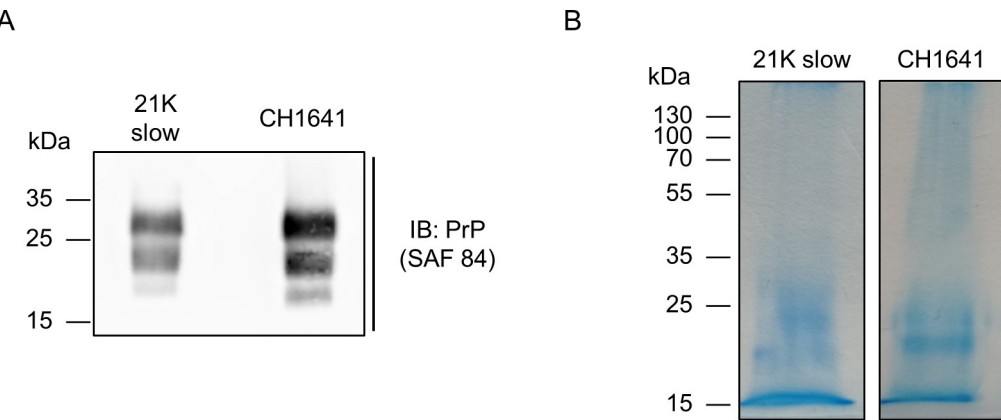

**Fig 1. Results of PrP^Sc isolation. A)** Immunoblot of PrP^Sc from 21K slow and CH1641 prion strains. **B)** Coomassie-stained gels of purified PrP^Sc 21K slow and CH1641 strains.

Wenborn *et al.* (2015) [32], including centrifugation steps, and PrP^Sc precipitation with sodium phosphotungstic acid (NaPTA) followed by digestion with PK.

21K slow [33–35] and CH1641 [36–39] are two prion substrains known to co-exist in sheep brain. The main way to differentiate them is biochemically; by digesting a sample composed of 21K slow prion strain, the unglycosylated band is 21 kDa, while CH1641 strain gives a 19 kDa unglycosylated band (seen in Fig 1A). Besides the biochemical differences, these strains differ also phenotypically. For example, histoblots from transgenic mice show different deposition patterns; in mice infected with 21K slow strain, PrP^res is present in habenula, thalamus, and hypothalamus, while in CH1641 infected mice, PrP^res is accumulated in the thalamus, oriens layer of the hippocampus, and cerebral cortex [34].

Both prion strains were successfully isolated, the obtained pellets were loaded on SDS-PAGE and stained with Coomassie (Fig 1B). To achieve the highest amount possible of each glycoform, only the diglycosylated bands from both strains were excised from the gel. The bands were cut into small pieces, transferred to tubes, and subjected to reduction and alkylation of disulfide bridges. Samples were then digested overnight with trypsin, by covering gel pieces with enough trypsin buffer, making sure they were completely covered by trypsin buffer. After peptide and glycopeptide extraction, samples were analyzed on LC-MS/MS. The analysis of each glycosylation site will be shown and discussed in further sections.

## LC-MS/MS analysis

PeptideMass [40] gave a list of theoretical masses of prion tryptic digests, which were in accordance with experimental data. The digestion of the prion protein resulted in two tryptic glycopeptides carrying glycosylation sites, differing in their masses and sequence: the first glycosylation site (N-184), located on the peptide YPNQVYYRPVDQYSNQNNFVHDCV^184-NITVK, with a mass of 3573.6735 Da (increased by 57 Da due to iodoacetamide treatment), and the second glycosylation site (N-200), located on a smaller peptide GE^200NFTETDIK, with a mass of 1152.5299 Da. The two peptides will further be referred to as simply N-184 and N-200.

Following trypsin digestion, dried glycopeptides were reconstituted in $H_2O$, and 10% of the total sample was loaded directly on LC-MS/MS. The resulting data were first analyzed with MaxQuant [41], a software that allows the identification and quantification of the detected peptides in the sample. Prion protein sequence coverage was 32% and 37.5% for CH1641 and

21K slow prion strain, respectively. The relative intensity corresponding to PrP[Sc] was 42.6% and 66.6% for 21K slow and CH1641, respectively. Other proteins detected were trypsin (the second most intense in both strains); ferritin, apolipoprotein E, and calcium/calmodulin-dependent protein kinase alpha type II, known as major prion contaminants [42–44], although contributing to less than 5% of the total intensity.

After confirming the presence of the prion protein, prion glycopeptides were searched for and annotated manually. Extracted ion chromatograms (EICs) of the MS/MS data were created for diagnostic glycan fragments, [HexNAcHex + H]$^+$, [HexNAc + H]$^+$, and [NeuAc + H]$^+$ (m/z values of 366.13 (1+), 204.08 (1+) and 292.09 (1+), respectively). The most abundant glycopeptides from each potential elution cluster were crosschecked with m/z values of prion glycopeptides from the internal database. If the match was found, MS/MS data of each analyte were annotated manually using Compass DataAnalysis (Bruker Daltonik GmbH), GlycoWorkbench [45], and GlycoMod [46]. The sequence of the peptide part and the composition of the attached N-glycan structure were confirmed by observed fragment ions.

However, by directly loading the samples, only N-184 glycopeptide was detected. Since glycopeptide ionization is usually suppressed by other coeluted peptides, this could be the reason why the N-200 glycopeptide was not detected. To overcome this issue glycopeptides were enriched with the hydrophilic interaction liquid chromatography-solid phase extraction (HILIC-SPE), a simple and rapid method used for small sample amounts [47]. By analyzing HILIC-SPE enriched samples, only N-200 glycopeptide was detected, most likely due to the differences in hydrophobicity of the two tryptic peptides. The relative quantitation was performed separately within one glycosylation site.

## 1.1 Qualitative and quantitative analysis of N-184 glycopeptides

In total, 50 different glycans were detected on the N-184 glycosylation site in both strains (the list presented in Fig 2). Out of those, 46 structures were mutual in both strains; therefore, they were taken into consideration for further comparison. The abundance of the remaining 4 structures, not taken for quantitation, was below 1%, meaning that there is a possibility they are present in both strains but were below detection limit.

The detected glycopeptides elute in a range from 33 to 36 min. The structures can be distinguished as neutral (18 structures), monosialylated (17 structures), disialylated (10 structures), and trisialylated (1 structure)–eluting in separate retention time windows: 33–33.8 min (neutral), 34–35.1 min (monosialylated), and 35.5–36.2 min (di- and trisialylated elute together, as seen in Fig 3A). All the detected structures were fucosylated, with at least one fucose residue present (either core or outer-arm fucose, which could not be distinguished for each structure because of the lack of MS/MS data). Most of the structures were tetraantennary (13 structures), 8 structures were biantennary, while there were 3 triantennary structures detected, and 16 structures carrying bisecting GlcNAc (Fig 3B, 3C and 3D). No oligomannose structures were detected, and the majority were complex type N-glycans, while only 4 structures were hybrid (Fig 3B and 3C).

The observed m/z values were either in 3+, 4+ and/or 5+ charged state, and all detected m/z values for each glycopeptide were used for generating the extracted ion chromatogram (EIC). Of the 46 structures, 20 of them were fragmented and contain corresponding MS/MS data. In case no MS/MS spectrum was obtained, the structure was reported based on the isotopic distribution pattern, the error between the theoretical and observed m/z values (structures with less than 20 ppm were taken into consideration), retention times of confirmed glycopeptides and monosaccharide mass difference from glycopeptides confirmed by MS/MS. In the MS/MS spectrum, glycan structural features were annotated with main oxonium ions: 204.08 (1+),

| Glycan composition | Proposed glycan structure | MS/MS | Theoretical glycopeptide m/z [M+H]+ | Observed glycopeptide m/z [M+4H]4+ | |
|---|---|---|---|---|---|
| | | | | 21K slow | CH1641 |
| H4N4F1 | | | 5181.2675 | 1296.0827 | 1296.0729 |
| H3N5F1 | | + | 5222.2941 | 1306.3342 | 1306.3257 |
| H5N3F2 | | | 5286.2988 | 1322.3471 | 1322.3376 |
| H4N4F2 | | | 5327.3254 | 1332.5862 | 1332.5796 |
| H5N4F1 | | | 5343.3203 | 1336.5882 | 1336.5800 |
| H4N5F1 | | | 5384.3469 | 1346.8484 | 1346.8311 |
| H5N3S1F1 | | | 5431.3363 | 1358.5942 | 1358.5791 |
| H4N4S1F1 | | | 5472.3629 | 1368.8400 | 1368.8356 |
| H5N4F2 | | + | 5489.3782 | 1373.1066 | 1373.0981 |
| H4N5F2 | | + | 5530.4048 | 1383.3618 | 1383.3519 |
| H4N6F1 | | | 5587.4263 | N/D | 1397.6054 |
| H5N4S1F1 | | | 5634.4157 | 1409.3585 | 1409.3594 |
| H5N4F3 | | | 5635.4361 | 1409.6158 | 1409.6079 |
| H4N5S1F1 | | + | 5675.4423 | 1419.6215 | 1419.6072 |
| H5N5F2 | | | 5692.4576 | 1423.8747 | 1423.8663 |
| H4N6F2 | | + | 5733.4842 | 1434.1335 | 1434.1218 |
| H5N4S1F2 | | | 5780.4736 | 1445.8700 | 1445.8629 |
| H6N4F3 | | | 5797.4889 | 1450.1328 | 1450.1278 |
| H4N5S1F2 | | | 5821.5002 | 1456.1256 | 1456.1211 |

| Glycan composition | Proposed glycan structure | MS/MS | Theoretical glycopeptide m/z [M+H]+ | Observed glycopeptide m/z [M+4H]4+ | |
|---|---|---|---|---|---|
| | | | | 21K slow | CH1641 |
| H5N5S1F2 | | + | 5983.5530 | 1496.6523 | 1496.6374 |
| H6N5F3 | | | 6000.5683 | 1500.8868 | 1500.8910 |
| H5N3S3F1 | | + | 6013.5271 | N/D | 1504.1230 |
| H5N6S1F1 | | | 6040.5745 | 1510.8963 | 1510.8949 |
| H5N6F3 | | + | 6041.5949 | 1511.1594 | 1511.1472 |
| H5N4S2F2 | | + | 6071.5690 | 1518.6426 | 1518.6335 |
| H6N5S1F2 | | | 6145.6058 | 1537.1672 | 1537.1539 |
| H6N5F4 | | | 6146.6262 | 1537.3997 | 1537.4007 |
| H5N6S1F2 | | + | 6186.6324 | 1547.4161 | 1547.4070 |
| H5N5S2F2 | | + | 6274.6484 | 1569.4168 | 1569.4053 |
| H6N5S1F3 | | | 6291.6637 | 1573.6724 | 1573.6595 |
| H5N6S2F1 | | + | 6331.6699 | 1583.6729 | 1583.6555 |
| H5N6S1F3 | | | 6332.6903 | 1583.9213 | 1583.9207 |
| H6N6F4 | | | 6349.7056 | 1588.1754 | 1588.1676 |
| H5N7S1F2 | | | 6389.7118 | 1598.1807 | 1598.1749 |
| H5N6S2F2 | | + | 6477.7278 | 1620.1879 | 1620.1741 |

**Fig 2. Detected N-184 glycopeptides on both prion strains.** Proposed glycan structures found on N-184 glycosylation site with the theoretical and observed m/z values of the detected glycopeptides. H–hexose, N–N-acetylhexosamine, F–fucose and S–N-acetylneuraminic acid (sialic acid). Blue square–N-acetylglucosamine (GlcNAc), green circle–mannose (Man), red triangle–fucose (Fuc), yellow circle–galactose (Gal), purple diamond–N-acetylneuraminic acid (Neu5Ac). The presence of MS/MS spectrum is indicated with +. N/D–not determined.

representing [HexNAc + H]+ residue; 366.13 (+1) [HexNAcHex + H]+; 512.19 (1+), in case of antennary fucosylation, representing [HexHexNAcFuc + H]+; and 657.22 (1+), in case of sialylation, [HexNAcHexNeuAc + H]+. The peptide portion was confirmed by the presence of b and y ions, corresponding to the amino acid sequence of the N-184 peptide, together with the fragment of the peptide backbone alone 1192.22 (3+), the peptide carrying GlcNAc residue 1259.92 (3+) or carrying core fucose (GlcNAc + Fuc), with m/z value of 1308.60 (3+) (seen in Fig 4).

One of the more interesting features was found on the N-glycan with a composition of H5N5S3F2. MS/MS data of the peptide carrying this glycan revealed the presence of a fragment with a value of 495.18 (1+), suggesting the presence of 6-sialyl-LewicC (6sLeC) structure [HexNAcNeuAc + H]+, where NeuAc residue is directly attached to GlcNAc residue, a

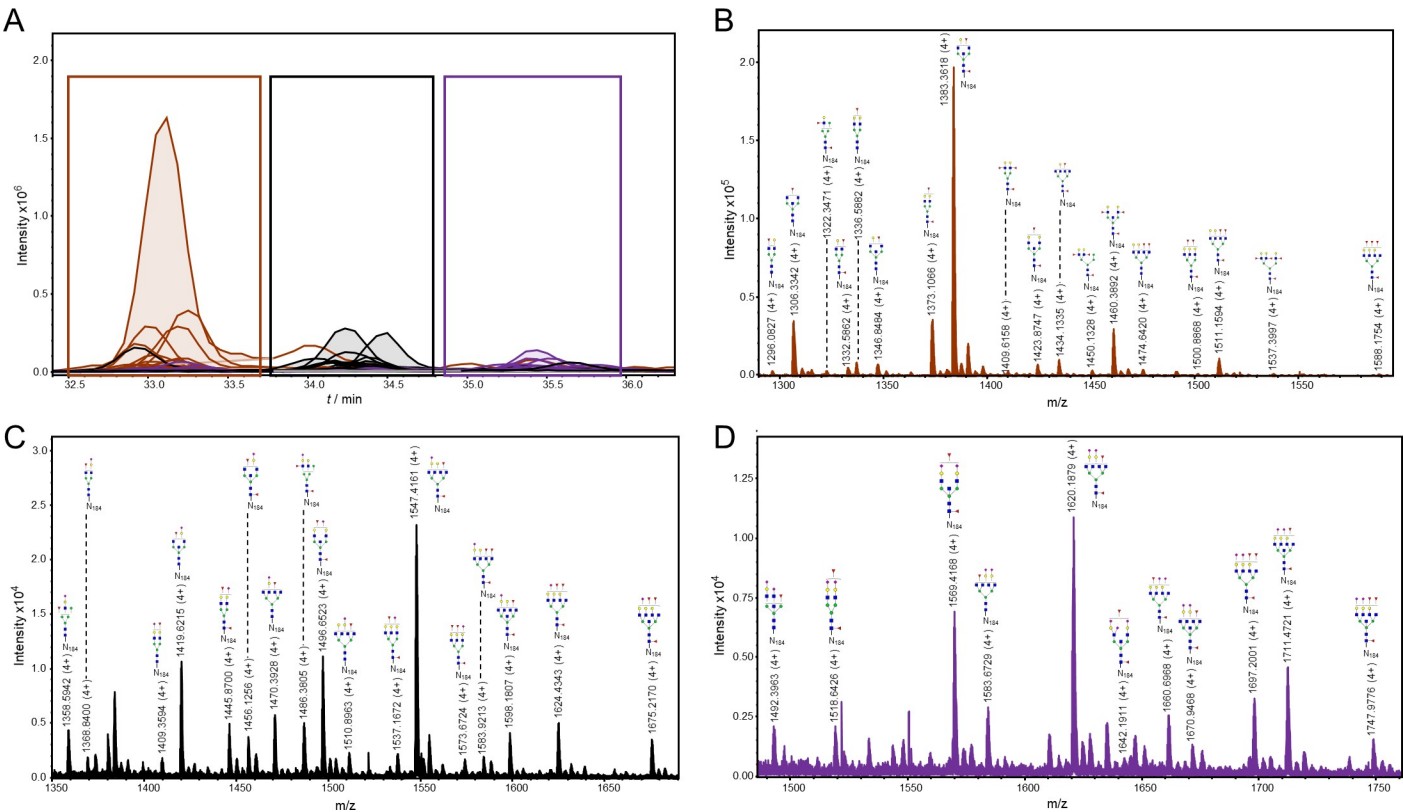

**Fig 3. Representation of detected glycoforms on the N-184 glycosylation site of 21K slow prion strain. A)** Extracted ion chromatograms with mutual 46 glycoforms detected. **B)** Assigned glycoforms in MS spectrum with N-184 peptide backbone: neutral, **C)** monosialylated and **D)** disialylated (together with one trisialylated glycoform). Blue square–N-acetylglucosamine (GlcNAc), green circle–mannose (Man), red triangle–fucose (Fuc), yellow circle–galactose (Gal), purple diamond–N-acetylneuraminic acid (Neu5Ac).

structure known to be present in mouse brain [48–50], although not reported previously on the mouse prion protein [28]. The presence of this feature should be further confirmed with orthogonal analytical methods.

The quantitation of all the mutual structures is represented in Fig 5, as relative abundances for each glycopeptide in each strain. N-glycan with a composition H4N5F2 was the most abundant one in both strains (contributing to around 30%), and interestingly, it is the same major structure found on mouse PrP$^{Sc}$ [28]. The next most abundant glycans detected were neutral structures: H3N5F1–7.46% and 6.69%, H5N4F2–5.92, and 5.18%, H5N5F3–5.63% and 5.91%, and a monosialylated structure H5N6S1F2–5.42 and 6.65%, abundances corresponding to 21K slow and CH1641 strain, respectively. The abundances of all other glycoforms were below 5%. Three structures found in CH1641 prion strain were not detected in 21K slow strain and one found in 21K slow strain was not detected in CH1641 strain. There is a possibility, however, that these structures were not detected in both strains due to their low abundance, since in the case of all the structures, it was less than 1.1%. Therefore, we can suggest that there are no major differences detected between the strains in the composition of the N-184 glycosylation site.

## Qualitative and quantitative analysis of N-200 glycopeptides

In total, 50 structures were found on the N-200 glycosylation site (summarized in Fig 6), out of which, 35 were mutual for both strains, which were used for analysis. The abundance of the

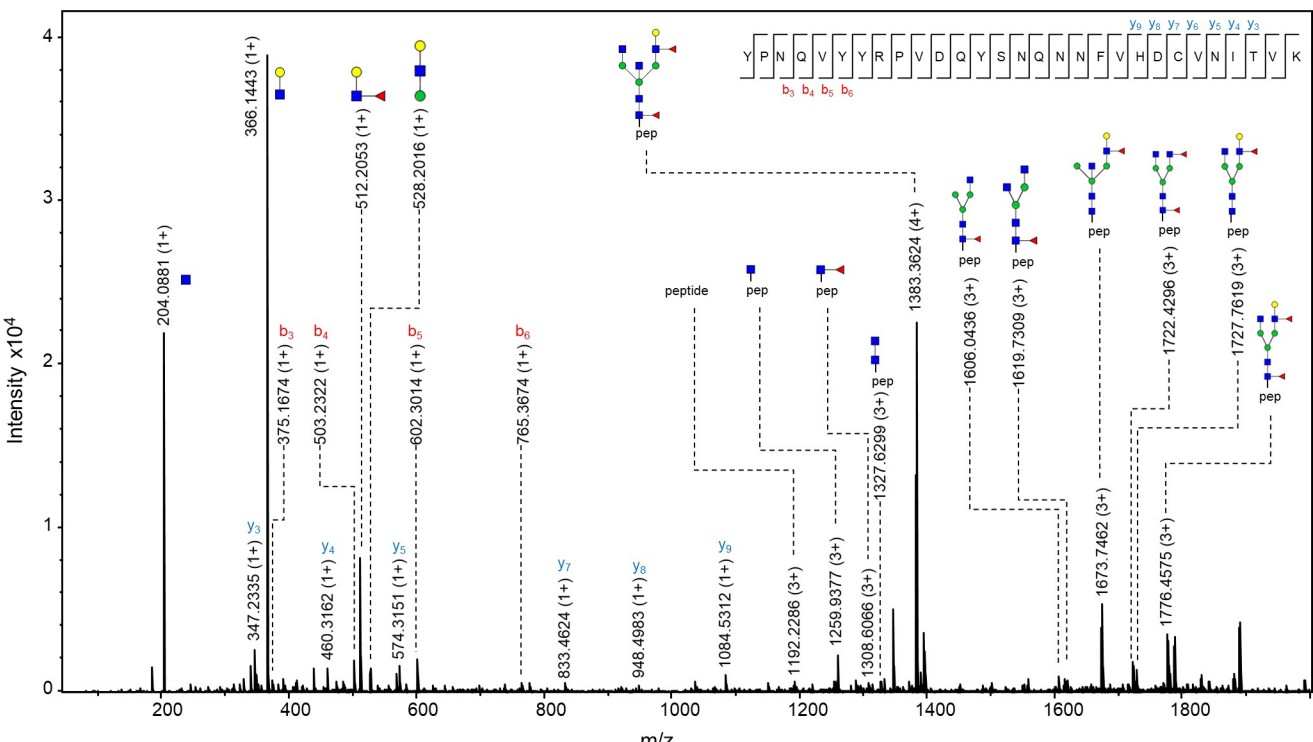

**Fig 4. MS/MS spectrum of N-184 glycopeptide from 21K slow prion strain.** The glycopeptide represented in the figure is the one carrying N-glycan with a composition H4N5F2, which was found to be the most abundant structure on this glycosylation site. Fragments of glycan-specific marker ions are represented, together with b and y ions confirming the amino acid sequence of N-184 peptide, and fragments of the peptide backbone with the N-glycan. Blue square–*N*-acetylglucosamine (GlcNAc), green circle–mannose (Man), red triangle–fucose (Fuc), yellow circle–galactose (Gal), purple diamond–*N*-acetylneuraminic acid (Neu5Ac).

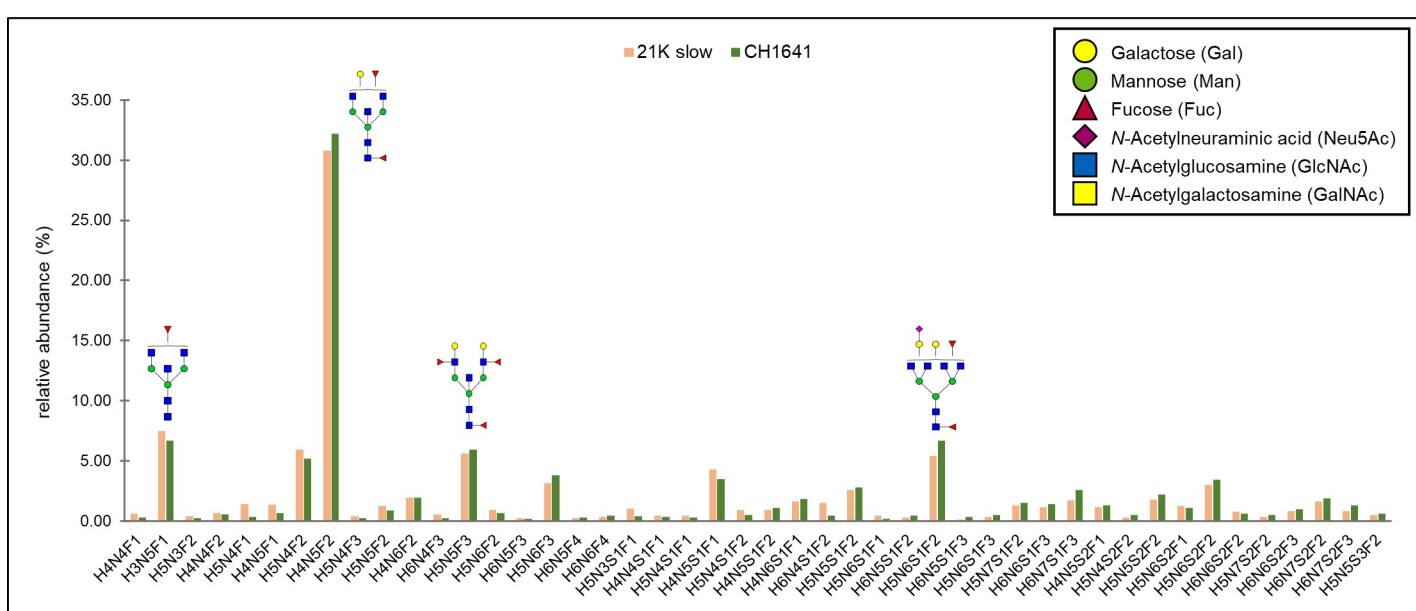

**Fig 5. Relative abundances of N-glycans on N-184 peptide in both strains.** In total, 46 structures are compared between the two strains analyzed. The most abundant structures (with relative abundance above 5%) are represented in the graph. Monosaccharide symbols marked in legend.

| Glycan composition | Proposed glycan structure | MS/MS | Theoretical glycopeptide m/z [M+H]+ | Observed glycopeptide m/z [M+3H]3+ | |
|---|---|---|---|---|---|
| | | | | 21K slow | CH1641 |
| H4N5F2 | | + | 3109.2612 | 1037.0770 | 1037.0940 |
| H4N5S1F1 | | + | 3254.2987 | 1085.4209 | N/D |
| H5N5F3 | | + | 3417.3719 | 1139.7785 | 1139.7950 |
| H4N6S1F1 | | | 3457.3781 | 1153.1143 | 1153.1314 |
| H5N6F2 | | | 3474.3934 | 1158.7851 | 1158.8039 |
| H5N5S1F2 | | + | 3562.4094 | 1188.1256 | 1188.1410 |
| H5N6S1F1 | | + | 3619.4309 | 1207.1327 | N/D |
| H8N3S1F2 | | + | 3642.4090 | 1214.7780 | N/D |
| H5N5S2F1 | | | 3707.4469 | 1236.4719 | N/D |
| H5N6S1F2 | | + | 3765.4888 | 1255.8162 | 1255.8294 |
| H6N6F3 | | + | 3782.5041 | 1261.4883 | 1261.5058 |
| H5N6S2F1 | | | 3910.5263 | 1304.1638 | 1304.1716 |
| H6N6S1F2 | | | 3927.5416 | 1309.8281 | 1309.8461 |
| H5N7S1F2 | | | 3968.5682 | 1323.5062 | 1323.5217 |
| H6N7F3 | | | 3985.5835 | N/D | 1329.1930 |
| H5N6S2F2 | | + | 4056.5842 | 1352.8481 | 1352.8607 |
| H6N6S1F3 | | + | 4073.5995 | 1358.5149 | 1358.5302 |
| H5N6S3F2 | | | 4347.6796 | 1449.8742 | 1449.8833 |
| H6N6S3F1 | | | 4363.6745 | 1455.2037 | N/D |
| H6N6S2F3 | | + | 4364.6949 | 1455.5453 | 1455.5596 |
| H5N7S3F1 | | | 4404.7011 | 1468.8783 | 1468.8859 |
| H6N7S2F2 | | + | 4421.7164 | 1474.5547 | 1474.5664 |
| H7N7S1F3 | | | 4438.7317 | 1480.2252 | 1480.2544 |
| H9N4S2F3 | | + | 4444.6945 | 1482.1989 | 1482.2092 |
| H6N6S3F2 | | + | 4509.7324 | 1503.8917 | 1503.9071 |
| H6N7S3F1 | | + | 4566.7539 | 1522.8985 | 1522.9029 |
| H6N7S2F3 | | | 4567.7743 | 1523.2372 | 1523.2523 |
| H7N7S2F2 | | | 4583.7692 | 1528.5692 | N/D |
| H7N7S1F4 | | | 4584.7896 | 1528.9110 | 1528.9252 |
| H6N6S3F3 | | + | 4655.7903 | 1552.5777 | 1552.5892 |
| H6N7S3F2 | | + | 4712.8118 | 1571.5834 | 1571.5939 |
| H7N7S2F3 | | + | 4729.8271 | 1577.2513 | 1577.2705 |

**Fig 6. Identified N-200 glycopeptides on both prion strains.** Proposed glycan structures found on N-200 glycosylation site with the theoretical and observed m/z values of the detected glycopeptides. H–hexose, N–N-acetylhexosamine, F–fucose and S–N-acetylneuraminic acid (sialic acid). Blue square–N-acetylglucosamine (GlcNAc), green circle–mannose (Man), red triangle–fucose (Fuc), yellow circle–galactose (Gal), purple diamond–N-acetylneuraminic acid (Neu5Ac). The presence of MS/MS spectrum is indicated with +. N/D–not determined.

remaining 15 structures was mostly below 2% (for only 2 structures detected on 21K slow strain, the abundance was above the mentioned value: 3.33% for H4N5S1F1 and 2.04% for H6N6S3F1). Therefore, we cannot exclude the possibility that the remaining structures are not present in each of the strains. The fact they were not detected could be due to the lower amount of the starting material or could be due to a lower amount of the prion protein in the CH1641 prion strain.

The N-200 glycopeptides elute from 19 to 26 min (Fig 7A). The glycoforms again elute in separate retention time windows, although some EICs are overlapping. This is due to sialylated structures eluting in two or three peaks, which is caused by isomers with different linkages (α2–3 or α2–6) and different antennae occupancy of sialic acid residues. [51]. From the 35 structures, only 4 are neutral, and 31 sialylated: 10 monosialylated, 11 disialylated, and 10

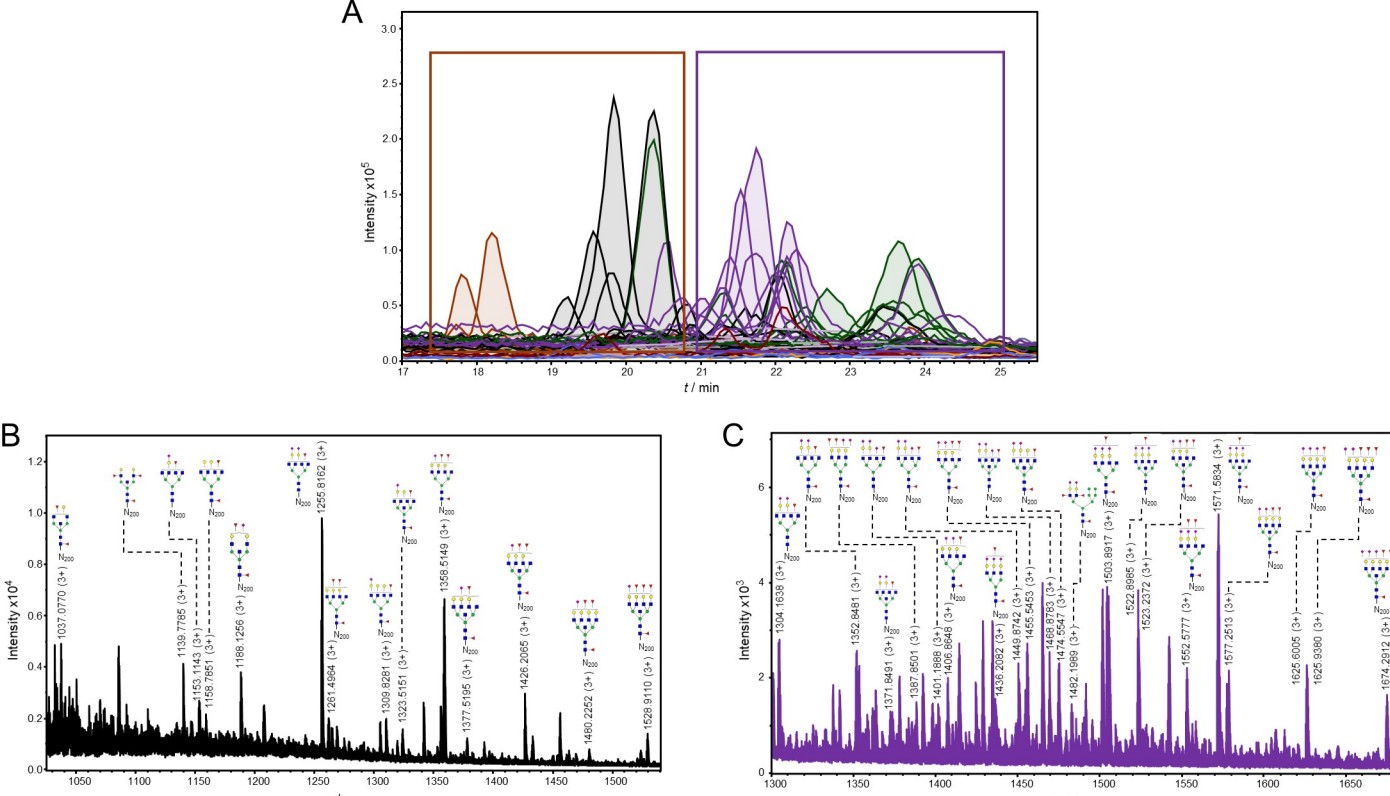

**Fig 7. Representation of detected glycoforms on the N-200 glycosylation site of 21K slow prion strain. A)** Extracted ion chromatograms with mutual 35 glycoforms detected. **B)** Assigned glycoforms in MS spectrum with N-200 peptide backbone: neutral and monosialylated, **C)** disialylated and trisialylated glycoforms. Blue square–*N*-acetylglucosamine (GlcNAc), green circle–mannose (Man), red triangle–fucose (Fuc), yellow circle–galactose (Gal), purple diamond–*N*-acetylneuraminic acid (Neu5Ac).

trisialylated. Again, all structures were fucosylated, around half of them had bisecting GlcNAc (18 structures), 14 were tetraantennary, only 2 were triantennary, and no biantennary structures were detected. Therefore, most of the structures were complex, while only one was hybrid (Fig 7B and 7C).

The observed m/z values of N-200 glycopeptides were in 2+, 3+, and/or 4+ charge state, and again all m/z values detected for each glycopeptide were used for creating EIC. From the 35 proposed structures, more than half of them (21 glycan composition) were supported by MS/MS spectra. The glycan portion was confirmed by the already above mentioned oxonium ions, while the peptide was confirmed by the presence of mainly y ions (and in some cases b ions) of the amino acids corresponding to N-200 peptide, together with the peptide backbone fragment with an m/z value of 1153.52 (1+), peptide carrying one GlcNAc residue 1356.58 (1+) and carrying core fucose 1559.68 (1+). Fig 8 represents the MS/MS spectrum of a glycopeptide carrying the glycan with a composition of H5N6S2F2, where again, the presence of 6sLeC feature was implied (fragment with m/z value of 495.18 (1+), corresponding to [HexNAc-NeuAc + H]$^+$ residue).

Relative abundances for each of the mutual N-200 glycopeptide in each strain is shown in Fig 9. The 12 most abundant structures are the same in both strains, just differing slightly in their ranking order, for instance: the most abundant structure for CH1641 strain was H6N7S2F2 (9.65%), which is the second most abundant in 21K slow (7.57%), where the most abundant one is H5N6S1F2 (8.63%). The relative abundance was above 5% for only 6

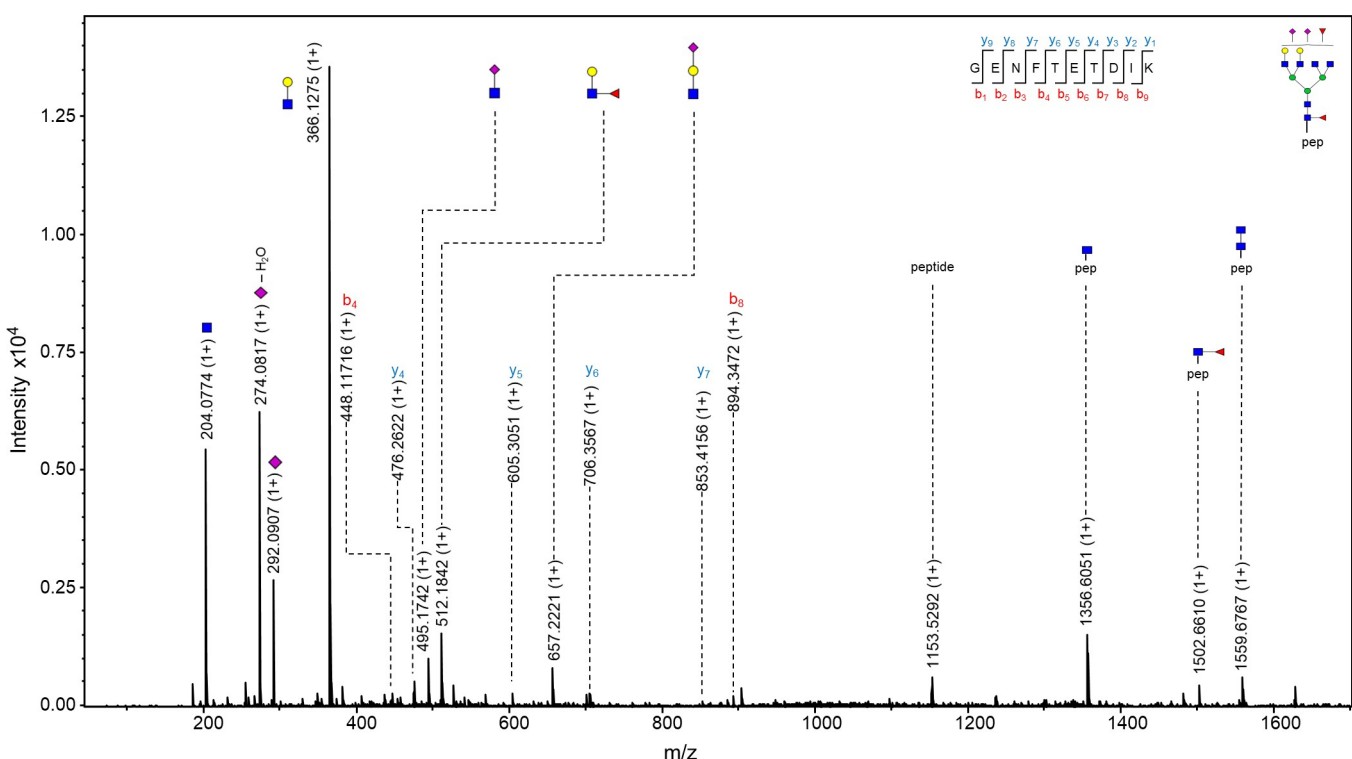

**Fig 8. MS/MS spectrum of N-200 glycopeptide from 21K slow prion strain.** The glycopeptide represented in the figure is the one carrying N-glycan with a composition H5N6S2F2. The usual fragments of glycan-specific marker ions are represented, additionally with a fragment of m/z value 495.1742 (1+), suggesting the presence of 6sLeC. Also, y ions confirming the amino acid sequence of the N-200 peptide and fragments of the peptide backbone with the N-glycan are visible. Blue square–*N*-acetylglucosamine (GlcNAc), green circle–mannose (Man), red triangle–fucose (Fuc), yellow circle–galactose (Gal), purple diamond–*N*-acetylneuraminic acid (Neu5Ac).

structures in the case of 21K slow, and 5 structures in the case of CH1641 strain, while the abundance was equal or below 1% for 9 structures in both strains. It is difficult to claim that these differences in relative abundance are due to strain differences or due to the integration itself, since some signals were low in intensity. Also, the slight differences in relative abundance are around 1–2%, which are negligible, therefore, we suggest that no major differences are detected neither in the N-200 glycosylation site between these two strains.

## Discussion

The precise mechanism of the conformational change of PrP$^C$ to PrP$^{Sc}$ is still unknown and remains one of the main issues to be solved in the prion field. The "protein-only" hypothesis is widely accepted, however, it fails to explain the large variety and existence of prion strains [52]. The strain phenomenon occurs in all species susceptible to TSEs, and it is of great importance to comprehend it since the public health risk of prion adaptation and transmission is still unknown [11,52–54]. The source of prion strain variation is partially due to PrP polymorphisms, which are present in most species, for example, sheep, goats [12,55], mice [56,57], and humans [58,59]. Human prion diseases are influenced by the polymorphism at codon 129 (where either methionine or valine can be present) [15], however, it is known that with just methionine at this position, there is an occurrence of different PrP strains [52], meaning that other factors may play a role in determining strain diversity.

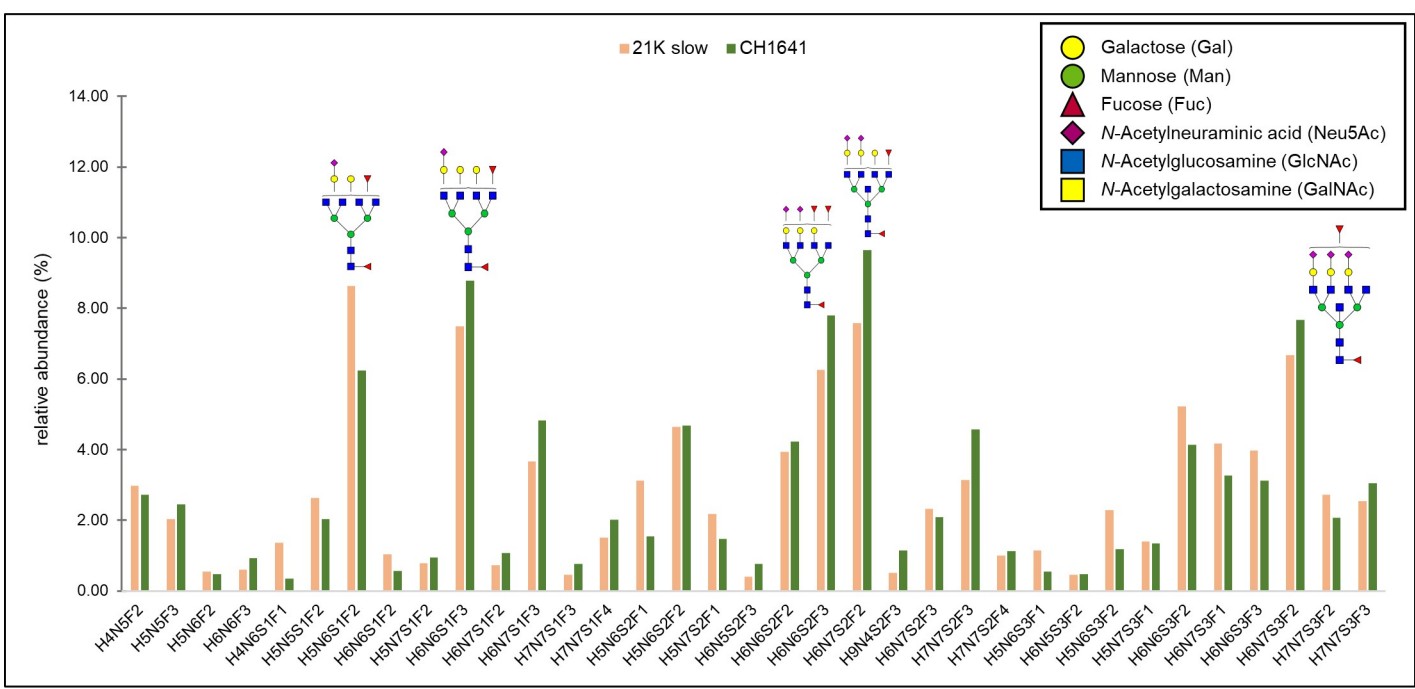

**Fig 9. Relative abundances of N-glycans on N-200 peptide.** In total, 35 structures are compared between the two strains analyzed. Structures with relative abundance above 5% are represented in the graph. Monosaccharide symbols marked in legend.

The idea that strain diversity is due to distinct conformational states in the prion protein backbone has widely been accepted [54]. One of the studied aspects, thought to have a role in the infectivity of PrP, is the glycosylation of the prion protein, although conflicting results have been reported throughout the years. For example, Cancellotti *et al.* (2013) obtained PrP from transgenic mice carrying mutations in one or both glycosylation sites (N-180 and N-196 in mice). Wild-type mice were intracerebrally inoculated with the mutated PrP, and in the case of some strains, properties were maintained in the host. However, there have been cases where new strains emerge, which led the researchers to the conclusion that glycans are not essential for prion replication, even though strain properties can be modified by changes in glycosylation [60]. In another study, transgenic mice were infected (carrying mutations at glycosylation sites) with various PrP strains and reported differences in susceptibility to prion disease. Transgenic mice carrying a mutation at the first (N-180) or both sites were almost completely resistant to the disease, while the absence of the second glycosylation site (N-196) enabled the transmission of the disease. This evidence implies that prion transmission can be influenced by changing the glycosylation status of PrP$^C$ in the host [61]. In contrast, by inoculating mice intraperitoneally, the absence of glycans slowed down or prevented the disease, suggesting that glycosylation has a role in prion replication, or affects the transport of prions from the peripheral tissue to the central nervous system (CNS) [62]. Another interesting study showing glycans are not crucial for prion propagation was the replication of mammalian prions in a transgenic invertebrate host. Unlike vertebrates, N-glycans in *Drosophila* are predominantly high mannose and paucimannose, and while the hybrid and complex are found, they are present in relatively low amounts [63,64]. Interestingly, Thackray *et al.* (2018) showed that biological properties of different sheep prion strains were conserved after propagation in the transgenic *Drosophila*, even though the glycosylation is significantly different between vertebrates and invertebrates [65]. *In vitro* studies, using protein misfolding cyclic amplification

(PMCA), have further confirmed that glycans are not essential for the conversion of PrP$^C$ to PrP$^{Sc}$ [66]. It has been proposed that glycans stabilize the structure of the protein, rendering PrP$^C$ more resistant to misfolding and interfering in PrP$^C$ and PrP$^{Sc}$ interaction [67]. Another study, using desialylated PrP$^C$, led to a proposal that PrP$^{Sc}$ amplification is increased because of the removal of electrostatic repulsion between N-glycans [68]. Although prion conversion is possible without glycans, the importance of investigating glycosylation in strains has been emphasized in a recent study, in which it was shown that strain conformation determines both cofactor and glycosylation preferences, supporting the strain-specific neurotropism hypothesis [69]. The same group of researchers has also shown *in vitro* that the second glycosylation site affects prion conversion, suggesting this site has a role in controlling the conversion of PrP$^{Sc}$ to PrP$^C$ [70]. Although studies investigating prion glycosylation have not always been consistent, they have led to the suggestion that the prion strain properties are encoded in the backbone of the protein.

However, the previously mentioned studies lack the information regarding exact glycan structures. Changes in N-glycosylation have been known to exist in a large variety of diseases [71], and in order to investigate and clarify the roles of N-glycans, a detailed characterization of specific glycan structures is necessary. So far, due to the complexity glycan analysis imposes, only a few studies have engaged to tackle this particular problem. Some studies have reported a specific subset of brain oligosaccharides, such as structures having a higher content of fucose and bisecting GlcNAc than reported usually in the brain [26–28,72], the same specifics of the subset was also observed in our analysis. A glycosyltransferase, called *N*-acetylglucosaminyl-transferase III (GnT-III) is responsible for the addition of bisecting GlcNAc residue in N-glycans. Generally, the presence of the bisecting GlcNAc is known to affect the activity of other glycosyltransferases, usually suppressing the synthesis of other epitopes in N-glycans [73]. Although the physiological functions of bisecting GlcNAc remain unclear, it has been shown that increasing its expression levels has a protective role in Alzheimer's disease [74]. Also, a possible involvement of this enzyme has been indicated in prion diseases, suggesting that there is a decrease in the activity of the enzyme in cells where PrP$^{Sc}$ is formed [27]. Additional studies are necessary to clarify the involvement and role of this specific residue, leading to answers that could potentially serve as a therapy treatment.

So far, the characterization of N-glycans has been performed on single prion strains, usually on mouse and hamster PrP$^{Sc}$. Here we report site-specific analysis of N-glycans on ovine PrP$^{Sc}$. This study also represents a comparison of glycan profiles on two different prion protein strains. First, the analysis revealed a great difference in the N-glycan composition between the two glycosylation sites, to a larger extent than reported before [28]. All the structures were fucosylated, and a large number of them had a bisecting GlcNAc residue, while the majority of structures were also sialylated. Interestingly, as also reported previously, no high mannose structures were detected on the prion protein, in contrast to the large abundance of those structures in brain tissue [75]. The neutral structure containing bisecting GlcNAc (with a composition of H4N5F2) is the most abundant structure found on both strains on N-184, contributing to around 30% of all structures. On the N-200 glycosylation site, major structures contributed to a maximum of 9.6% of the relative abundance, meaning there is no prevailing structure like on the N-184 site. However, this site was shown generally to be heavily sialylated, when compared to N-184 glycosylation site, the same trend reported previously on mouse PrP$^{Sc}$ [28]. The role of sialic acid in prions has been increasingly investigated in recent years. For example, it has been shown that sialic acid is a part of the "self-associated molecular pattern" (SAMP) [76]. By removing sialic acid from the surface of glycans, galactose becomes exposed, which sends a signal to macrophages [77] or leads to phagocytosis of neurons [78], suggesting that the fate of prions is affected by their sialylation status [79]. Sialic acid also has

an effect on the species barrier, eliminating it by reducing the sialylation status of the host PrP[C] [68]. Finally, a recent study has shown that the infectivity of PrP[Sc] can be influenced by its sialylation status; by removing sialic acids from PrP[Sc], the infectivity is lost and by restoring the sialylation it is switched on again [80]. Another additional interesting feature found on both glycosylation sites was the presence of 6sLeC structure, known to occur in mice brains [50], although not many glycoproteins with this structure have been reported.

Our data report additional information on the PrP glycan composition, previously not reported on mouse or hamster PrP[Sc] [26–28,72]. In total, 100 glycan structures were detected on both strains. The main point observed in this work is the fact that the glycan composition in different strains does not show major differences, suggesting, at least in these strains analyzed, that the strain properties could indeed be encoded in the protein, and not the glycans themselves. Other studies have confirmed that the presence of glycans is not crucial for prion diseases to occur and that the glycosylation of host PrP[C] influences glycosylation of newly formed PrP[Sc] [16,81]. Since glycosylation varies in different cells and brain regions [82], it has been proposed that prion strains preferentially accumulate in those cells (or distinct brain regions) where the specific glycoform of PrP[C] is located [83]. Glycosylation is known to have a role in many diseases; specifically, site occupancy is correlated with congenital disorders of glycosylation (CDG) [84]. Therefore, a better understanding of glycosylation is important to diagnose and develop possible treatments for different diseases.

Here we report the analysis of the diglycosylated bands, showing no major differences in the glycosylation profile between the two strains. However, in both strains, the diglycosylated band is the dominant one, seen in the immunoblot (Fig 1B), which could also be the explanation of the similar glycan profile. For future studies, the monoglycosylated PrP[Sc] band remains to be investigated, to determine site occupancy in these two strains, and its possible role. Also, it would be interesting to expand this type of analysis to other strains and organisms, especially the ones displaying differences in their immunoblot pattern; for example, to compare glycan profiles of strains where the diglycosylated band is more dominant (e.g. classical BSE strain) to ones where the monoglycosylated is more dominant (e.g. L-type BSE strain).

## Materials and methods

### Ethics statement

All animal experiments were performed in compliance with institutional and French national guidelines and in accordance with the European Directives 86/609/EEC and 2010/63/EU. The animal experiments that are part of this study (national registration 01734.01) were approved by the local ethics committee (Comité d'éthique locale de l'Ecole Nationale Vétérinaire de Toulouse).

### Scrapie brain material

Scrapie material was obtained by endpoint titration of one passage of scrapie in ovine VRQ (tg338) transgenic mice. Two scrapie strains, 21K slow and CH1641, were further transmitted to sheep by intracerebral inoculation. Both strains were produced in Cheviot sheep with VRQ/VRQ genotype; the animals were age-matched and kept under the same feeding and environmental conditions. After the propagation of scrapie in sheep, strain typing was performed for each sheep brain isolate by back transmission in tg338 mice to make sure there was no strain divergence between the original and propagated prions.

Brain tissue was collected from euthanized sheep. A total of two isolates weighing 20 g was used for 21K slow strain, and three isolates weighing 10 g for CH1641 strain. Further on, using a glass tissue homogenizer grinder, 20% (w/v) brain tissue homogenates were prepared for

each isolate in 1X phosphate-buffered saline (PBS, pH 7.4) containing protease inhibitor cocktail (cOmplete, EDTA-free Protease Inhibitor Cocktail, Roche) and 4% N-lauroylsarcosine sodium salt (sarkosyl, Sigma). Brain homogenates were centrifuged at 1,000 g for 10 minutes at room temperature in a bench centrifuge (Eppendorf), and 15 mL of aliquots of the supernatants (SNs) were transferred to clean 50 mL tubes (Falcon) and stored at -80˚C until PrP$^{Sc}$ purification. In the end, a total of 130 mL and 150 mL of 20% brain homogenate was used for 21K slow and CH1641 strain, respectively.

## PrP$^{Sc}$ isolation and purification

Isolation was performed based on a protocol described previously [32]. Briefly, iodixanol (OptiPrep Density Gradient Medium, Sigma) was added to 15 mL of brain homogenate, so that the final concentration reached 10%. Samples were centrifuged in a bench centrifuge (Eppendorf) for 30 min at 4,500 g, after which they were divided into a pellet (P1) and supernatant (SN1). All the SN1 (for each strain from all the isolates) were transferred to clean 1.5 mL tubes (Eppendorf) and centrifuged in a bench microfuge (Eppendorf) at 5,000 g for 1 h. Again, each sample was divided into a pellet (P2) and supernatant (SN2). All SN2 were transferred into clean 1.5 mL tubes, and sodium phosphotungstic acid (NaPTA, Sigma) was added to reach the final concentration of 0.5%. The sample was incubated with constant shaking at 500 rpm for 1 h, at 37˚C and then centrifuged at 25,000 g for 90 min in an ultracentrifuge (Beckman Coulter). The pellets (P3) were collected, sonicated in 2% sarkosyl containing 0.5% PTA, and incubated with shaking for 30 min. The samples were centrifuged again at 25,000 g for 90 min, after which three of the obtained pellets (P4) were pooled together and sonicated in 1X RIPA buffer (25 mM Tris HCl pH 8.0, 150 mM NaCl, 1% Nonidet P-40 substitute, 1% sodium deoxycholate). The samples were digested with 20 μg/mL of proteinase K (PK, Roche) and incubated with constant shaking at 400 rpm at 37˚C for 30 min. The reaction was stopped with 2 mM of phenylmethylsulphonyl fluoride (PMSF, Sigma), and the samples were centrifuged at 186,000 g for 1 h, at 6˚C in an ultracentrifuge. The supernatants were discarded, and the obtained pellets (P5) were pooled together (12 of them), sonicated in 1X RIPA, incubated with shaking at 1,000 rpm for 15 min, at room temperature. The pellet obtained after centrifugation at 186,000 g was collected and rewashed with 1X RIPA. The final pellet (P7), for each strain, consisted of pooled PrP$^{Sc}$–isolated from all brain homogenate prepared (130 mL in case of 21K slow and 150 mL for CH1641 strain). Each P7 was resuspended in 2X sample loading buffer (125 mM Tris-HCl pH 6.8, 200 mM dithiothreitol (DTT), 20% glycerol, 10% 2-mercaptoethanol, 4% SDS, and 0.2% bromphenol blue), boiled at 100˚C for 10 min, and used further for SDS-PAGE or immunoblotting.

## SDS-PAGE, immunoblotting and Coomassie staining

Samples were loaded onto a 12% Tris-Glycine SDS-PAGE gel. PageRuler Plus Prestained Protein Ladder (Thermo Fisher Scientific) was used as a molecular weight marker. The gel electrophoresis was performed at 90 V for approximately 30 min, then at a constant voltage of 120 V until the dye front ran off the end of the gel. Gels were used for immunoblotting or Coomassie staining.

Proteins were transferred onto nitrocellulose membranes (GE Healthcare) for 120 min at 250 mA by Criterion Blotter (Bio-Rad). The membranes were blocked with 5% non-fat milk in TBS-T (200 mM Tris, 1.5 mM NaCl, and 0.05% Tween-20) for 1 h, after which they were incubated overnight at 4˚C with anti-PrP SAF 84 antibody diluted 1:500 in the blocking solution. Membranes were washed three times with TBS-T, after which they were incubated with horseradish peroxidase (HRP)-conjugated secondary antibody goat anti-mouse diluted 1:1000 in

blocking solution. After three washes again with TBS-T, the signal was detected using enhanced chemiluminescence (GE Healthcare), and the band intensity was acquired using the UVI Soft software (Uvitec Alliance, Cambridge).

Gels were also stained in GelCode Blue Safe Protein Stain (Thermo Fisher Scientific) by gently shaking for 1 h. Gels were destained overnight in ultrapure water and used further for in-gel trypsin digestion.

### In-gel trypsin digestion and glycopeptide enrichment

The protocol was performed as described previously [85]. Briefly, bands corresponding to diglycosylated PrP$^{Sc}$ were excised with a clean scalpel and cut into smaller cubes. The gel pieces were transferred to 1.5 mL microcentrifuge tubes and reduced with 10 mM of dithiothreitol (DTT, Sigma) for 30 min at 56˚C. After cooling down, samples were alkylated with 55 mM iodoacetamide (IAA, Sigma) for 20 min at room temperature. Gel pieces were destained by alternating incubation with 100 mM ammonium bicarbonate/ACN (1:1, v/v) and neat ACN. Samples were digested with 6.5 ng/μL of trypsin (Promega) overnight at 37˚C. Tryptic digests were extracted by shaking in 5% formic acid/ACN (1:2, v/v). Extraction was repeated two times, and the digests were pooled and dried in a vacuum concentrator.

Glycopeptide enrichment was performed using solid-phase extraction on Chromabond HILIC beads (Macherey-Nagel). The beads were added to wells of a filter plate (Orochem), the samples were loaded onto beads in 80% ACN containing 0.1% TFA (v/v) and washed two times with the same solvent. All the washes were removed by vacuum filtration (Pall). Elution was performed with 0.1% TFA, after which the glycopeptides were dried in a vacuum concentrator and reconstituted in 20 μL of ultrapure water.

### LC-ESI-MS/MS analysis of prion glycopeptides

Digested glycopeptides were separated on nanoAcquity chromatographic system (Waters, Milford, MA) coupled to Compact mass spectrometer (Bruker, Bremen, Germany) with an electrospray ionization (ESI) source. Samples were loaded either directly after the overnight trypsin digestion (2 μL from 20 μL) or after the enrichment procedure (20 μL). They were loaded onto a PepMap 100 C18 trap column (5 mm x 300 μm, Thermo Fisher Scientific) at a flow rate of 40 μL/min of solvent A (0.1% formic acid) to wash off impurities and salts. Glycopeptides were separated on C18 analytical column (150 mm x 100 μm, 100 Å, Advanced Materials Technology) using a linear gradient from 0% to 80% of solvent B (80% ACN) in solvent A, at a flow rate of 1 μL/min in a 90-minute analytical run.

Fragmentation of glycopeptides was performed by tandem MS/MS by using CaptiveSpray interface, where nanoBooster was used to introduce gaseous acetonitrile into nitrogen flow. The mass spectrometer operated in positive ion mode; capillary voltage was set to 1300 V, nitrogen pressure was set to 0.2 bar, and the drying gas to 4.0 l/min at 150˚C. Auto MS/MS method was used by selecting three precursor ions and exclusion criteria after three MS/MS spectra. Mass range was from 50 m/z to 4000 m/z, with a spectra rate of 1 Hz. Transfer time was from 70 μs to 150 μs, and pre-pulse storage was 12 μs.

### Data processing

The sheep PrP amino acid sequence was obtained from the UniProt database (accession number Q712V9 for sheep with VRQ/VRQ polymorphism). PeptideMass [40] was used to obtain theoretical masses of trypsin digested peptides. MaxQuant [41] was used to identify and quantify all the peptides present in the sample, with methionine oxidation and N-terminus acetylation as variable modifications, and cysteine carbamidomethylation as a fixed modification.

The identification of glycopeptides was performed manually using DataAnalysis software (version 4.4, Bruker).

Extracted ion chromatogram (EIC) of the MS/MS data for the m/z value 366.13 (1+), representing [HexNAcHex + H]$^+$ was created and each sample was searched for the presence of the glycan oxonium ions, such as the aforementioned, together with 204.08 (1+), representing [HexNAc + H]$^+$ and in case of sialylation 292.09 (1+), representing [NeuAc + H]$^+$. Glyco-Workbench [45] was used to display glycan structures, also calculating theoretical masses of the glycopeptide fragments, which were searched for in the MS/MS spectra. After confirming the presence of prion glycopeptides at a given retention time, the base peak chromatogram (BPC) was searched for other m/z values, and a list of all multiply charged ions was created, corresponding to potential prion glycopeptides. The values were converted to singly charged ions and characterized using GlycoMod [46], proposing all the possible glycan compositions. After determining all the possible glycopeptides, EIC was created for all the compositions, by including m/z values of all multiply charged states.

All EICs were integrated using DataAnalysis, and relative abundances were normalized by total area by dividing the peak area of each glycopeptide in each sample with the total chromatographic area of the sample.

## Acknowledgments

We thank Jelena Šimunović for suggestions and advice given during analysis.

## Author Contributions

**Conceptualization:** Gordan Lauc, Giuseppe Legname.

**Data curation:** Natali Nakić, Thanh Hoa Tran.

**Formal analysis:** Natali Nakić, Thanh Hoa Tran, Mislav Novokmet.

**Investigation:** Natali Nakić, Thanh Hoa Tran.

**Methodology:** Natali Nakić, Thanh Hoa Tran, Mislav Novokmet.

**Resources:** Olivier Andreoletti, Gordan Lauc, Giuseppe Legname.

**Supervision:** Gordan Lauc, Giuseppe Legname.

**Writing – original draft:** Natali Nakić, Thanh Hoa Tran, Giuseppe Legname.

**Writing – review & editing:** Natali Nakić, Thanh Hoa Tran, Mislav Novokmet, Olivier Andreoletti, Gordan Lauc, Giuseppe Legname.

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
