## [Decision Letter · Decision Letter 0]

26 Oct 2020

Dear Prof. Legname,

Thank you very much for submitting your manuscript "Site-specific analysis of N-glycans from different sheep prion strains" for consideration at PLOS Pathogens. As with all papers reviewed by the journal, your manuscript was reviewed by members of the editorial board and by several independent reviewers. The reviewers appreciated the attention to an important topic. Based on the reviews, we are likely to accept this manuscript for publication, providing that you modify the manuscript according to the review recommendations.

Sincerely,

Surachai Supattapone

Associate Editor

PLOS Pathogens

Neil Mabbott

Section Editor

PLOS Pathogens

Kasturi Haldar

Editor-in-Chief

PLOS Pathogens

orcid.org/0000-0001-5065-158X

Michael Malim

Editor-in-Chief

PLOS Pathogens

orcid.org/0000-0002-7699-2064

Reviewer Comments (if any, and for reference):

Reviewer's Responses to Questions

**Part I - Summary**

Reviewer #1: This paper analyses the glycan structure of two ovine prion strains. Little is known about the specific glycan structures of prions and most studies have concentrated on murine models to date. However the glycosylation status of prion strains has been thought to play a major role in disease transmission and pathogenesis. This study adds to our basic knowledge but also raises further questions and future research directions.

This study has concentrated on ovine strains which is novel compared to earlier studies on mouse and hamster strains and reports that no major differences have been identified between the two strains. However the study has concentrated on two strains which are predominantly diglycosylated and has only investigated the diglycosylated band. It is unclear whether differences would be apparent in either the monoglycosylated band or between strains which are predominantly monoglycosylated and this could be construed as a weakness of the current study.

Reviewer #2: This manuscript reports on a very complete descriptive study of the glycans present in two strains of sheep PrPSc. The main strength of the study is that the results are very solid and have been obtained with stat-of-the-art methodology. Its main limitation, that these results are mostly descriptive: while there are differences in the glycan composition of the two strains, the Authors do not use these data to address any biological issue (for example: different biological and/or pathogenic properties of these two strains).

Reviewer #3: Refolding of the cellular glycoprotein PrPC into the infectious conformer PrPSc leads to prion disease. PrPC is glycosylated at two asparagines via complex N-linked glycans. PrPSc is also glycosylated but very few studies have been done to characterize its N-glycans. Differences in the way PrPSc from different prion strains is glycosylated may help to explain differences in prion disease phenotypes. The study by Nakic et al characterizes the N-glycans in PrPSc from two distinct strains of sheep scrapie. They identify over 100 structures at both N-linked glycosylation sites in PrPSc but find no significant differences in N-glycan composition between the two strains. They conclude that PrPSc glycosylation does not account for the differences in prion strain phenotype.

This is a well-designed study which adds to the limited literature characterizing the N-linked glycan composition of prion protein. The data are clearly presented, support the conclusions drawn, and would be of interest to prion researchers.

**Part II – Major Issues: Key Experiments Required for Acceptance**

Reviewer #1: (No Response)

Reviewer #2: No major experimental or methodological issues have been identified by this reviewer.

Reviewer #3: (No Response)

**Part III – Minor Issues: Editorial and Data Presentation Modifications**

Reviewer #1: The science appears sound however I have a few minor comments.

Lines 130-142: Add reference for the two ovine strains and a brief description in how they differ or are similar. Even for a prion scientist, keeping track of strains can be difficult. Why were these two strains chosen? Why was only the diglycosylated band studied?

Lines 265 - 268: there is a comment about the amount of starting material. I'm a little unclear as to what the authors mean. If it's relative abundance, this should take into account different amounts of starting material. Is it possible to equalise starting material (I note they used 10% brain homogenates) or were any differences calculated using relative intensity on western blots?

Lines 305-312: more recent studies could be included here (they may be from after this manuscript was submitted), two papers by Burke et al., 2020 PloS Path).

Lines 333-334: The results centre around the differences between strains and this is nicely summarised in the results. Here the authors talk about great difference in composition between the two sites. Could a brief line or two summarising these differences be added to make it clearer to the reader what they are?

Lines 368-375: This does give a brief comment on factors raised above (Lines 130-142), could examples be given?

Reviewer #2: 1) Beginning in line 64, the Authors refer to the protein-only hypothesis. Currently the prion concept is overwhelmingly accepted, and therefore it would be more appropriate to indicate that prions are infectious proteins, or infectious particles composed of protein only. Otherwise, there is the danger that "the protein-only hypothesis" will become an equivocal fossilized term similar to the "theory" of Evolution.

2) The sentence beginning in line 67 is ambiguous. I suggest rewriting it as: ...laboratory. However, the one aspect of prion biology, the existence of strains, still needs to be fully elucidated.

3) The meaning of the paragraph beginning in line 71 is not clear, in part perhaps because of its odd syntax.

4) Given that many readers will not be familiar with the methodology used, it might be helpful to introduce in the Results section some additional summary methodological descriptions, for example, indicate "in-gel digestion", briefly summarize how the specific glycans are identified etc.

Reviewer #3: 1) Other than a difference in the size of PrPSc (Figure 1), the strains used are not described. It would be helpful for the authors to give a brief summary of how the two sheep strains analyzed, 21K slow and CH1642, differ phenotypically. For example, how does the pattern of PrPSc deposition in the brain differ between the two isolates? What are the clinical signs of disease in sheep and how do the incubation times differ?

2) The Methods mention that 5 isolates for 21K slow and 3 isolates for CH1641 were typed in transgenic mice. However, it is not clear if all of the isolates were used for the glycan analysis. Was each isolate analyzed individually or was material from all of the isolates for a given strain combined? If all of the isolates were analyzed individually, this increases the robustness of the dataset and greatly strengthens the conclusions. The authors need to clearly state how many isolates of each strain were included in the final analysis.

3) The authors also need to clarify in the Methods whether or not the PrPSc analyzed was derived from sheep brain or transgenic mouse brain. The Methods on line 389 states that brain tissue from transgenic mice was collected, but it’s not clear how many brains were collected or whether they were actually used for the N-glycan analysis.

4) In the Tables, please define N/D. Presumably it means “Not Detected” but this should be specified.

PLOS authors have the option to publish the peer review history of their article (what does this mean?). If published, this will include your full peer review and any attached files.

Reviewer #1: No

Reviewer #2: No

Reviewer #3: No
---

## [Decision Letter · Decision Letter 1]

10 Dec 2020

Dear Prof. Legname,

We are pleased to inform you that your manuscript 'Site-specific analysis of N-glycans from different sheep prion strains' has been provisionally accepted for publication in PLOS Pathogens.

Best regards,

Surachai Supattapone

Associate Editor

PLOS Pathogens

Neil Mabbott

Section Editor

PLOS Pathogens

Kasturi Haldar

Editor-in-Chief

PLOS Pathogens

orcid.org/0000-0001-5065-158X

Michael Malim

Editor-in-Chief

PLOS Pathogens

orcid.org/0000-0002-7699-2064

Reviewer Comments (if any, and for reference):

Reviewer's Responses to Questions

**Part I - Summary**

Reviewer #1: This is a revision of a previously submitted manuscript. The authors have answered the questions and queries of all the reviewers and have added additional information, citations and clarity to the manuscript. This has benefited the manuscript and it now very clear and detailed and will be of great use/interest to scientists in this area.

In this instance, carrying out more studies on monoglycosylated strains would be beyond the scope of the manuscript and likely funding/time constraints but this study gives proof that these laboratory studies are possible and lead the way to further in depth studies.

Reviewer #2: (No Response)

Reviewer #3: The authors have satisfactorily addressed the points raised in my original review.

**Part II – Major Issues: Key Experiments Required for Acceptance**

Reviewer #1: N/A

Reviewer #2: (No Response)

Reviewer #3: None.

**Part III – Minor Issues: Editorial and Data Presentation Modifications**

Reviewer #1: N/A

Reviewer #2: (No Response)

Reviewer #3: None.

PLOS authors have the option to publish the peer review history of their article (what does this mean?). If published, this will include your full peer review and any attached files.

Reviewer #1: No

Reviewer #2: No

Reviewer #3: No

---

## [Editor Report · Acceptance letter]

25 Jan 2021

Dear Prof. Legname,

We are delighted to inform you that your manuscript, "Site-specific analysis of N-glycans from different sheep prion strains," has been formally accepted for publication in PLOS Pathogens.

Best regards,

Kasturi Haldar

Editor-in-Chief

PLOS Pathogens

orcid.org/0000-0001-5065-158X

Michael Malim

Editor-in-Chief

PLOS Pathogens

orcid.org/0000-0002-7699-2064